# Safe Driving via Expert Guided Policy Optimization

**Zhenghao Peng**[†*]**, Quanyi Li**[§*]**, Chunxiao Liu**[‡]**, Bolei Zhou**[†]
[†] The Chinese University of Hong Kong, [‡] SenseTime Research,
[§] Centre for Perceptual and Interactive Intelligence

**Abstract:** When learning common skills like driving, beginners usually have domain experts standing by to ensure the safety of the learning process. We formulate such learning scheme under the Expert-in-the-loop Reinforcement Learning where a guardian is introduced to safeguard the exploration of the learning agent. While allowing the sufficient exploration in the uncertain environment, the guardian intervenes under dangerous situations and demonstrates the correct actions to avoid potential accidents. Thus ERL enables both exploration and expert's partial demonstration as two training sources. Following such a setting, we develop a novel Expert Guided Policy Optimization (EGPO) method which integrates the guardian in the loop of reinforcement learning. The guardian is composed of an expert policy to generate demonstration and a switch function to decide when to intervene. Particularly, a constrained optimization technique is used to tackle the trivial solution that the agent deliberately behaves dangerously to deceive the expert into taking over. Offline RL technique is further used to learn from the partial demonstration generated by the expert. Safe driving experiments show that our method achieves superior training and test-time safety, outperforms baselines with a substantial margin in sample efficiency, and preserves the generalizabiliy to unseen environments in test-time. Demo video and source code are available at: https://decisionforce.github.io/EGPO/.

**Keywords:** Safe Reinforcement Learning, Human-in-the-loop Machine Learning, Autonomous Driving

## 1 Introduction

Reinforcement Learning (RL) shows promising results in human-interactive applications ranging from autonomous driving [1], the power system in smart building [2], to the surgical robotics arm [3]. However, training and test time safety remains as a great concern for the real-world applications of RL. This problem draws significant attention since the agent needs to explore the environment sufficiently in order to optimize its behaviors. It might be inevitable for the agent to experience dangerous situations before it can learn how to avoid them [4], even the training algorithms contain sophisticated techniques to reduce the probability of failures [5, 6, 7].

We humans do not learn purely from trial-and-error exploration, for the sake of safety and efficiency. In daily life, when learning some common skills like driving, we usually ensure the safety by involving domain expert to safeguard the learning process. The expert not only demonstrates the correct actions but also acts as a *guardian* to allow our own safe exploration in the uncertain environment. For example as illustrated in Fig.1, when learning to drive, the student with the learner's permit can directly operate the vehicle in the driver's seat while the instructor stands by. When a risky situation happens, the instructor takes over the vehicle to avoid the potential accident. Thus the student can learn how to handle tough situations both from the exploration and the instructor's demonstrations.

In this work, we formulate such learning scheme with Expert-in-the-loop RL (ERL). As shown in the right panel of Fig.1, ERL incorporates a *guardian* in the interaction between agent and environment. The guardian contains a *switch* mechanism and an *expert* policy. The switch decides to intervene the free exploration of the agent in the situations when the agent is conducting unreasonable behaviors or

---

 [*] Zhenghao Peng and Quanyi Li contribute equally to this work.

5th Conference on Robot Learning (CoRL 2021), London, UK.

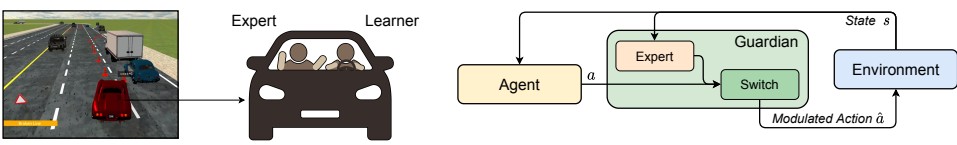

Figure 1: The expert intervenes the learner in dangerous situations. We model it through the Expert-in-the-loop RL scheme on the right panel where a guardian is introduced in the loop of the interaction between agent and environment.

a potential critical failure is happening. In those cases the expert takes over the main operation and starts providing demonstrations on solving the task or avoiding dangers. Our setting of ERL extends previous works of Human-in-the-loop RL in two ways: First, the guardian inspects the exploration all the time and actively intervenes if necessary, instead of passively advising which action is good [8] or evaluating the collected trajectories after the agent rolling out [9, 10]. This feature guarantees the safe exploration in training time. Second, the guardian does not merely intervene the exploration and terminate the episode [11], instead, it demonstrates to the agent the correct actions to escape risky states. Those demonstrations become effective training data to the agent.

Following the setting of ERL, we develop a novel method called *Expert Guided Policy Optimization (EGPO)*. EGPO addresses two challenges in ERL. First, the learning agent may abuse the guardian and consistently cause intervention so that it can exploit the high performance and safety of the expert. To tackle this issue, we impose the Lagrangian method on the policy optimization to limit the intervention frequency. Moreover, we apply the PID controller to update the Lagrangian multiplier, which substantially improves the dual optimization with off-policy RL algorithm. The second issue is the partial demonstration data collected from the guardian. Since those data is highly off-policy to the learning agent, we introduce offline RL technique into EGPO to stabilize the training with the off-policy partial demonstration. The experiments show that our method can achieve superior training safety while yielding a well-performing policy with high safety in the test time. Furthermore, our method exhibits better generalization performance compared to previous methods.

As a summary, the main contributions of this work are: (1) We formulate the Expert-in-the-loop RL (ERL) framework that incorporates the guardian as a demonstrator as well as a safety guardian. (2) We develop a novel ERL method called *Expert Guided Policy Optimization (EGPO)* with a practical implementation of guardian mechanism and learning pipeline. (3) Experiments show that our method achieves superior training and test safety, outperforms baselines with a large margin in sample efficiency, and generalizes well to unseen environments in test time.

## 2   Related Work

**Safe RL**. Learning RL policy under safety constraints [12, 13, 7] becomes an important topic in the community due to the safety concern in real-world applications. Many methods based on constrained optimization have been developed, such as the trust region methods [5], Lagrangian methods [5, 6, 14], barrier methods [15, 16], Lyapunov methods [4, 17], *etc*. Another direction is based on the safety critic, where an additional value estimator is learned to predict cost, apart from the primal critic estimating the discounted return [7, 18]. Saunders et al. [11] propose HIRL, a scheme for safe RL requiring extra manual efforts to demonstrate and train an imitation learning decider who intervenes the endangered agent. Differently, in our work the guardian does not terminate the exploration but instead continues the trajectory with the expert demonstrating the proper actions to escape risky states. However, majority of the aforementioned methods hold the issue that only the upper bound of failure probability of the learning agent can be guaranteed theoretically, but there is no mechanism to explicitly ensure the occurrence of the critical failures. Dalal et al. [19] assume that cost function is the linear transformation of the action and thus equip the policy network with a safety layer that can modulate the output action as an absolutely safe action. The proposed EGPO utilizes the guardian to ensure safe exploration without assuming the structure of the cost function.

**Learning from Demonstration**. Many works consider leveraging the collected demonstrations to improve policy. Behavior Cloning (BC) [20] and Inverse RL [21] uses supervised learning to fit the policy function or the reward function respectively to produce the same action as the expert. GAIL [22, 23, 24] and SQIL [25] ask the learning agent to execute in the environment and collect

trajectories to evaluate the divergence between the agent and the expert. This exposes the agent to possibly dangerous states. DAgger [26] periodically queries the expert for new demonstrations and is successfully applied to extensive domains [27, 28]. Recently, offline RL draws wide attention which learns policy from the dataset generated by arbitrary policies [29, 30, 31]. The main challenge of offline RL is the out-of-distribution (OOD) actions [30]. Conservative Q-Learning (CQL) [32] addresses the impact of OOD actions by learning a conservative Q-function to estimate the lower bounds of true Q values. In this work, we use CQL technique to improve the training on the trajectories with partial demonstrations given by the guardian.

**Human-in-the-loop RL**. An increasing number of works focus on incorporating human into the training loop of RL. The human is responsible for evaluating the trajectories sampled by the learning agent [9, 33, 10], or being a consultant to guide which action to take when the agent requests [8]. Besides, the human can also actively monitor the training process, such as deciding whether to terminate the episode if potential danger is happening [34, 11]. Human-Gated DAgger (HG-DAgger) [27] and Expert Intervention Learning (EIL) [35] utilize experts to intervene exploration and carry the agent to safe states before giving back the control. However, it is much less explored in previous works on how to (1) optimize the agent to minimize interventions, (2) efficiently utilize the data generated in free exploration and (3) learn from the takeover trajectories given by the expert. Addressing these aforementioned challenges, our work is derived from the Human-in-the-loop framework where the guardian plays the role of human expert to provide feedback to the learning agent.

# 3 Expert Guided Policy Optimization

Extending the setting of Human-in-the-loop RL, we frame the *Expert-in-the-loop RL* (ERL) that incorporates the guardian to ensure training safety and improve efficiency. We develop a novel method called *Expert Guided Policy Optimization (EGPO)* to implement the guardian mechanism.

## 3.1 Overview of the Guardian Mechanism

Taking learning to drive as a motivating example, generally speaking, the student driver learns the skills of driving from the instructor through two approaches: (1) *Student learns from instructor's demonstrations*. At the early stage of training, the student observes the demonstrations given by the instructor and learns rapidly by imitating the behaviors. Besides, the student also learns how the expert tackles dangerous situations; (2) *Student in driver's seat operates the vehicle in an exploratory way while the instructor serves as guardian.* The student can explore freely until the instructor conducts takeover of the vehicle in dangerous situations. Therefore, the student learns to drive from both the imitation of the expert and the free exploration.

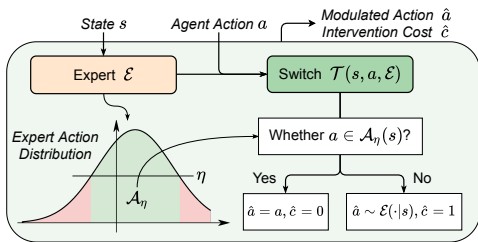

Figure 2: Flowchart of the guardian mechanism.

Based on this motivating example, we have the framework of *Expert-in-the-loop RL* (ERL). As illustrated in the right panel of Fig. 1, we introduce the component of *guardian* on top of the conventional RL scheme, which resembles the instructor who not only provides high-quality demonstrations to accelerate the learning, but also safeguards the exploration of agent in the environment. In the proposed EGPO method, the guardian is composed of two parts: an expert and a switch function.

The **expert** policy $\mathcal{E} : a^E \sim \mathcal{E}(\cdot|s)$ can output safe and reliable actions $a^E$ in most of the time. Besides, it can provide the probability of taking action $a$ produced by the agent: $\mathcal{E}(a|s) \in [0, 1]$. This probability reflects the agreement of the expert on the agent's action, which serves as an indicator for intervention in the switch function. We assume the access to such well-performing expert policy. The **switch** is another part of the guardian, which decides under what state and timing the expert should intervene and demonstrate the correct actions to the learning agent. As shown in Fig. 2, the switch function $\mathcal{T}$ considers the agent action as well as the expert and outputs the modulated action $\hat{a}$ fed to the environment and the intervention occurrence $\hat{c}$ indicating whether the guardian is taking over the

control:
$$\mathcal{T}(s, a, \mathcal{E}) = (\hat{a}, \hat{c}) = \begin{cases} (a, 0), & \text{if } a \in \mathcal{A}_\eta(s) \\ (a^E \sim \mathcal{E}(\cdot|s), 1), & \text{otherwise,} \end{cases} \tag{1}$$

wherein $\eta$ is the confidence level on the expert action probability and $\mathcal{A}_\eta(s) = \{a \in \mathcal{A} : \mathcal{E}(a|s) \geq \eta\}$ is the confident action space of the expert. The switch mechanism leads to the formal representation of the behavior policy:

$$\hat{\pi}(a|s) = \pi(a|s)\mathbf{1}_{a \in \mathcal{A}_\eta(s)} + \mathcal{E}(a|s)F(s), \tag{2}$$

wherein $F(s) = \int_{a' \notin \mathcal{A}_\eta(s)} \pi(a'|s)da'$ is a function denoting the probability of the agent choosing an action that will be rejected by the switch. Emulating how human drivers judge the risky situations, we rely on the expert's evaluation of the safety during training, instead of any external objective criterion.

We derive the guarantee on the training safety from the introduction of guardian. We first have the assumption on the expert:

**Assumption 1** (Failure probability of the expert). *For all state, the step-wise probability of expert producing unsafe action is bounded by a small value $\epsilon < 1$: $\mathbb{E}_{a \sim \mathcal{E}(\cdot|s)} I(s, a) \leq \epsilon$, wherein $I(s, a) \in \{0, 1\}$ is a Boolean denotes whether next state $s' \sim \mathcal{P}(s'|s, a)$ is an ground-truth unsafe state.*

We use the expected cumulative probability of failure to measure the expected risk encountered by the behavior policy: $\hat{V} = \mathbb{E}_{s_0} \hat{V}(s_0) = \mathbb{E}_{s_0, \tau \sim P(\hat{\pi})} \sum_{t=0} \gamma^t I(s_t, a_t)$ wherein $P(\hat{\pi})$ refers to the trajectory distribution deduced by the behavior policy. We propose the main theorem of this work:

**Theorem 1** (Upper bound of the training risk). *The expected cumulative probability of failure $\hat{V}$ of the behavior policy $\hat{\pi}$ in EGPO is bounded by the step-wise failure probability of the expert $\epsilon$ as well as the confidence level $\eta$:*

$$\hat{V} \leq \frac{\epsilon}{1 - \gamma}(1 + \frac{1}{\eta} + \frac{\gamma}{1 - \gamma}K'_\eta),$$

*wherein $K'_\eta = \max_s \int_{a \in \mathcal{A}_\eta(s)} da$ has **negative correlation** to $\eta$.*

When $\epsilon$ is fixed, increasing the confidence level will shrink the upper bound of $\hat{V}$, leading to better training safety. The proof is given in the Appendix.

In the implementation, the actions from agent are firstly modulated by the guardian and the safe actions will be applied to the environment. We update the learning agent with off-policy RL algorithm. Meanwhile, we also leverage a recent offline RL technique to address the partial demonstrations provided by the guardian and further improve the learning stability. The policy learning is presented in Sec. 3.2. Since the intervention from guardian indicates the agent has done something wrong, we also optimize the policy to reduce intervention frequency through the constrained optimization in Sec. 3.3.

## 3.2 Learning Policy from Exploration and Partial Demonstration

The proposed EGPO method can work with most of the RL algorithms to train the safe policy since the guardian mechanism does not impose any assumption on the underlying RL methods. In this work, we use an off-policy actor-critic method Soft Actor-Critic (SAC) [36] to train the agent. The method utilizes two neural networks including a Q network estimating the state-action value: $Q_\phi$, and a policy network: $\pi_\theta$. $\phi$ and $\theta$ are the parameters. The training algorithm alternates between the policy evaluation and the policy improvement in each iteration. The policy evaluation process updates the estimated Q function by minimizing the L2 norm of the entropy regularized TD error:

$$y(r_t, s_{t+1}) = r_t + \gamma \mathop{\mathbb{E}}_{a_{t+1} \sim \pi_\theta(\cdot|s_{t+1})} [Q_{\bar{\phi}}(s_{t+1}, a_{t+1}) - \alpha \log \pi_\theta(a_{t+1}|s_{t+1})],$$

$$L_Q(\phi) = \frac{1}{2} \mathop{\mathbb{E}}_{(s_t, a_t, r_t, s_{t+1}) \sim \mathcal{B}} [y(r_t, s_{t+1}) - Q_\phi(s_t, a_t)]^2. \tag{3}$$

Here $\mathcal{B}$ is the replay buffer, $\bar{\phi}$ is the delayed parameters, $\alpha$ is a temperature parameter. On the other hand, the policy improvement objective, which should be minimized, is written as:

$$L_\pi(\theta) = - \mathop{\mathbb{E}}_{s_t \sim \mathcal{B}, a_t \sim \pi_\theta(\cdot|s_t)} [Q_\phi(s_t, a_t) - \alpha \log \pi_\theta(a_t|s_t)]. \tag{4}$$

Since we use a safety-ensured mixed policy $\hat{\pi}$ to explore the environment, part of the collected transitions contain the actions from the expert. This part of data comes as *partial demonstration* denoted as $\mathcal{B}^{\mathcal{E}}$, which leads to the distributional shift problem. Many works have been proposed to overcome this problem, such as the V-trace in the on-policy algorithm IMPALA [37], the advantage-weighted actor-critic [38] in the off-policy algorithm, and many other offline RL methods [31, 30, 32]. To train with the off-policy data produced by the guardian, we adopt the recent Conservative Q-Learning (CQL) [32], known as an effective offline RL method, in our *Learning from Partial Demonstration (LfPD)* setting. The objective to update Q function becomes:

$$L_Q^{\text{LfPD}}(\phi) = \beta(\underbrace{\mathbb{E}_{s\sim\mathcal{B}^{\mathcal{E}},a\sim\pi_\theta}[Q_\phi(s,a)] - \underbrace{\mathbb{E}_{s\sim\mathcal{B}^{\mathcal{E}},a\sim\mathcal{E}}[Q_\phi(s,a)]}_{\text{2nd Term}}) + \underbrace{\frac{1}{2}\mathbb{E}_{(s,a)\sim\mathcal{B}}[y(r_t,s_{t+1}) - Q_\phi(s_t,a_t)]^2}_{\text{3rd Term}}.$$

$$\underbrace{\phantom{\mathbb{E}_{s\sim\mathcal{B}^{\mathcal{E}},a\sim\pi_\theta}}}_{\text{1st Term}}$$

$$(5)$$

Note that the 1st Term and 2nd Term are expectations over only the partial demonstration $\mathcal{B}^{\mathcal{E}}$, instead of the whole batch $\mathcal{B}$. In the partial demonstration data, the 1st Term reduces the Q values for the actions taken by the agent, while the 2nd Term increases the Q values of expert actions. The 3rd Term is the original TD learning objective in Eq. 3. CQL reflects such an idea: be conservative to the actions sampled by the agent, and be optimistic to the actions sampled by the expert. Minimizing Eq. 5 can lead to a better and more stable Q function. In next section, we discuss another hurdle in the training and propose a solution for intervention minimization.

## 3.3 Intervention Minimization via Constrained Optimization

The guardian intervenes the exploration of the agent once it behaves dangerously or inefficiently. However, if no measure is taken to limit intervention frequency, the learning policy is prone to heavily rely on the guardian. It deceives guardian mechanism by always taking dangerous actions so the guardian will take over all the time. In this case, the learning policy receives high reward under the supervision of guardian but fails to finish tasks independently.

In this section, we consider the intervention minimization as a constrained optimization problem and apply the Lagrangian method into the policy improvement process. Concretely, the optimization problem becomes: $\theta^* = \arg\max_\theta \mathbb{E}_{\pi_\theta}[\sum_{t=0}\gamma^t r_t]$, s.t. $\mathbb{E}_{\pi_\theta}[\sum_{t=0}\gamma^t\hat{c}_t] \leq C$ wherein $C$ is the intervention frequency limit in one episode. The Lagrangian dual form of the above problem becomes an unconstrained optimization problem with a penalty term:

$$\theta^* = \arg\max_\theta \min_{\lambda\geq 0} \mathbb{E}_{\tau\sim\hat{\pi}}\{(\sum_{t=0}\gamma^t r_t) - \lambda[(\sum_{t=0}\gamma^t\hat{c}_t) - C]\}, \tag{6}$$

where $\lambda \geq 0$ is known as the Lagrangian multiplier. The optimization over $\theta$ and $\lambda$ can be conducted iteratively between policy gradient ascent and stochastic gradient descent (SGD).

We additionally introduce an intervention critic $Q_\psi^C$ to estimate the cumulative intervention occurrence $\sum_{t'=t}\gamma^{(t-t')}\hat{c}_{t'}$. This network can be optimized following Eq. 3 with the reward replaced by the intervention occurrence. intervention minimization objective $L_\pi^\lambda$ can be written as:

$$L_\pi^\lambda(\theta) = \mathbb{E}_{s_t\sim\mathcal{B},a_t\sim\pi_\theta(\cdot|s_t)}[Q_\psi^C(s_t,a_t) - C], \tag{7}$$

Now we can update the policy by combining the policy improvement objective Eq. 4 with the intervention minimization objective Eq. 7 to the final objective:

$$L_\pi'(\theta) = L_\pi(\theta) + \lambda L_\pi^\lambda(\theta). \tag{8}$$

Conducting SGD on Eq. 8 w.r.t. $\theta$ can improve the return while reduce the intervention.

The SAC with the Lagrangian method has been proposed by Ha et al. [39]. From the attempt to reproduce the result in our task, we find that directly optimizing the Lagrangian dual in the off-policy RL algorithm SAC is highly unstable. Stooke et al. [6] analyze that optimizing Lagrangian multiplier brings oscillations and overshoot, which destabilizes the policy learning. This is because the update of the multiplier is an integral control from the perspective of control theory. Introducing the extra proportional and derivative control to update the Lagrangian multiplier can reduce the oscillations and

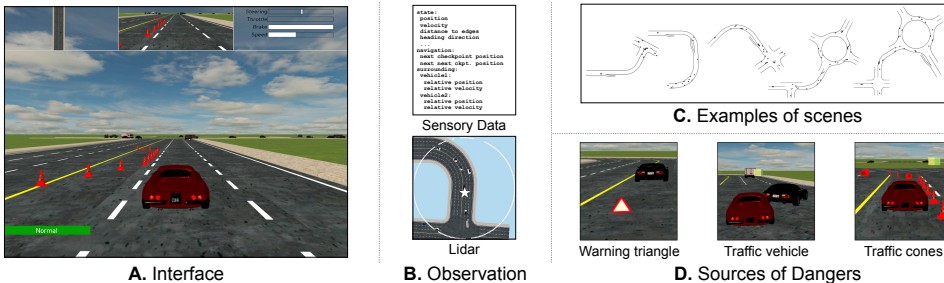

**A. Interface**    **B. Observation**    **C. Examples of scenes**    **D. Sources of Dangers**

Figure 3: **A**. The interface of the environment from MetaDrive [40]. **B**. The observations feeding to the target vehicle. **C**. The examples of the scenes we use in training and test. **D**. The three events creating costs: crashing with warning triangle, cone or other vehicles. $+1$ cost is given once those events occur.

corresponding cost violations. We thus adopt a PID controller to update $\lambda$ and form the responsive intervention minimization as:

$$\lambda \leftarrow K_p \delta + K_i \int_{i=1}^{k} \delta di + K_d \frac{\delta}{di}, \quad \text{wherein} \quad \delta = \mathbb{E}_{\tau}[\sum_{t=0} \hat{c}_t] - C, \tag{9}$$

where we denote the training iteration as $i$, and $K_p$, $K_i$, $K_d$ are the hyper-parameters. Optimizing $\lambda$ with Eq. 6 reduces to the proportional term in Eq. 9, while the integral and derivative terms compensate the accumulated error and overshoot in the intervention occurrence. We apply the PID controller in EGPO, as well as the baseline SAC-Lagrangian method in the experiments. Empirical results validate that PID control on $\lambda$ brings stabler learning and robustness to hyperparameter.

## 4 Experiments

### 4.1 Experimental Settings

**Environment**. We evaluate the proposed method and baselines in the recent driving simulator MetaDrive [40]. The environment supports generating an unlimited number of scenes via the Procedural Generation. Each of the scenes includes the vehicle agent, the complex road network, the dense traffic flow, and many obstacles such as cones and warning triangles, as shown in Fig. 3**D**. The task for the agent is to steer the target vehicle with low-level signals, namely acceleration, brake and steering, to reach the predefined destination. Each collision to the traffic vehicles or obstacles yields $+1$ environmental cost. The episodic cost in test time is the measurement on the safety of a policy, which is independent to whether the expert is used during training. The reward function only contains a dense driving reward and a sparse terminal reward. The dense reward is the longitudinal movement toward destination in Frenet coordinates. The sparse reward $+20$ is given when the agent arrives the destination. We build our testing benchmark based on MetaDrive rather than other RL environments like the safety gym [41] because we target on the application of autonomous driving and the generalization of the RL methods. Different to those environments, MetaDrive can generate an infinite number of driving scenes which allows evaluating the generalization of different methods by splitting the training and test sets in the context of safe RL.

**Split of training and test sets**. Different from the conventional RL setting where the agent is trained and tested in the same fixed environment, we focus on evaluating the generalization through testing performance. We split the scenes into the training set and test set with 100 and 50 different scenes respectively. At the beginning of each episode, a scene in the training or test set is randomly selected. After each training iteration, we roll out the learning agent *without guardian* in the test environments and record the percentage of successful episodes over multiple evaluation episodes, called *success rate*. Besides, we also record the episodic cost given by the environment and present it in following tables and figures.

**Training expert policy**. In our experiment, the expert policy is a stochastic policy trained from the Lagrangian PPO [41] with batch size as large as 160,000 and a long training time. To further improve the performance of the expert, we have reward engineering by doubling the cost and adding complex penalty to dangerous actions.

Table 1: The test performance of different approaches.

| Category | Method | Episodic Return | Episodic Cost | Success Rate |
|---|---|---|---|---|
| *Expert* | *PPO-Lag* | *392.38 ±99.47* | *1.26±0.57* | *0.86±0.05* |
| RL | SAC-RS | 346.49 ±16.51 | 8.68 ±3.34 | 0.68 ±0.10 |
|  | PPO-RS | 294.10 ±22.28 | 3.93 ±4.19 | 0.41 ±0.09 |
| Safe RL | SAC-Lag | 333.90 ±19.00 | 2.21 ±1.08 | 0.65 ±0.14 |
|  | PPO-Lag | 288.04 ±53.72 | 1.03 ±0.34 | 0.43 ±0.21 |
|  | CPO | 194.06 ±108.86 | 1.71 ±1.02 | 0.21 ±0.29 |
| Offline RL | CQL | 373.95 ±8.89 | 0.24 ±0.30 | 0.72 ±0.11 |
| IL | BC | 362.18 ±6.39 | **0.13 ±0.17** | 0.57 ±0.12 |
|  | Dagger | 346.16 ±22.62 | 0.67 ±0.23 | 0.66 ±0.12 |
|  | GAIL | 309.66 ±12.47 | 0.68 ±0.20 | 0.60 ±0.07 |
| Ours | EGPO | **388.37 ±10.01** | 0.56 ±0.35 | **0.85 ±0.05** |

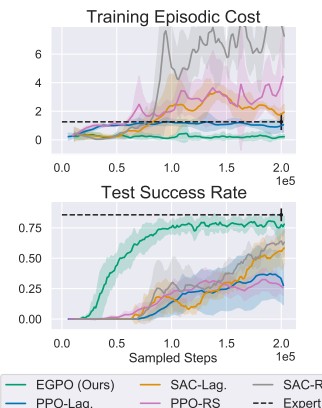

Figure 4: Comparison between our method and safe RL baselines.

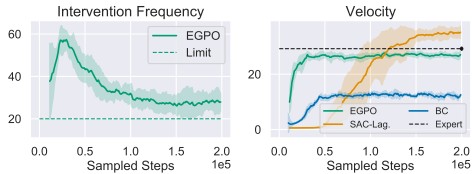
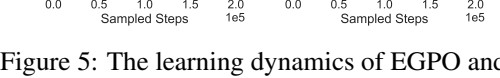

Figure 5: The learning dynamics of EGPO and baseline methods during training.

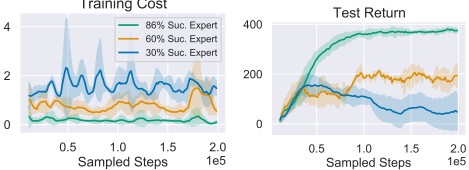

Figure 6: The curves of EGPO agents when varying the quality of experts.

**Implementation details**. We conduct experiments using RLLib [42], a distributed learning system which allows large-scale parallel experiments. Generally, we host 8 concurrent trials in an Nvidia GeForce RTX 2080 Ti GPU. Each trial consumes 2 CPUs with 8 parallel rollout workers. Each trial is trained over roughly $200,000$ environmental steps, which corresponds to 11 hours of individual driving experience. All experiments are repeated 5 times with different random seeds. Information about other hyper-parameters is given in Appendix.

## 4.2 Results

**Compared to RL and Safe RL baselines**. We evaluate two RL baselines PPO [43] and SAC [36] with the reward shaping (RS) method that considers negative cost as auxiliary reward. We also evaluate three safe RL methods, namely the Lagrangian version of PPO and SAC [6, 39] and Constrained Policy Optimization (CPO) [5]. As shown in Fig. 4 and Table 1, EGPO shows superior training and test time safety compared to the baselines. During training, EGPO limits the occurrence of dangers, denoted by the episodic cost, to almost zero. Noticeably, EGPO achieves lower cost compared to the expert policy. EGPO also learns rapidly and results to a high test success rate.

**Compared to Imitation Learning and Offline RL baselines**. We use the expert to generate $250,000$ steps of transitions from training environments and use this dataset to train with Behavior Cloning (BC), GAIL [22], DAgger [26], and offline RL method CQL [32]. As shown in Table 1, EGPO yields better test time success rate compared to the imitation learning baselines. BC outperforms ours in test time safety, but we find that BC agent learns conservative behaviors resulting in poor success rate and low average velocity to 15.05 km/h, while EGPO runs normally in 27.52 km/h, as shown in Fig. 5.

**Learning dynamics**. We denote the intervention frequency by the average episodic intervention occurrence $\mathbb{E}_\tau \sum_{t=0} \hat{c}_t$. As illustrated in Fig. 5, at the beginning of the training, the guardian is involved more frequently to provide driving demonstrations and prevent agent from entering dangerous states. After acquiring primary driving skills, the agent is prone to choosing actions that are more acceptable by guardian and thus the takeover frequency decreases.

Table 2: The test performance when ablating components in EGPO.

| Experiment | Episodic Return | Episodic Cost | Success Rate |
|---|---|---|---|
| **(a)** W/ rule-based switch | $339.10_{\pm 11.41}$ | $0.91_{\pm 0.60}$ | $0.57_{\pm 0.09}$ |
| **(b)** W/o intervention min. | $38.31_{\pm 3.61}$ | $1.00_{\pm 0.00}$ | $0.00_{\pm 0.00}$ |
| **(c)** W/o PID in SAC-Lag. | $338.80_{\pm 16.23}$ | $0.59_{\pm 0.40}$ | $0.67_{\pm 0.10}$ |
| **(d)** W/o CQL loss | $378.00_{\pm 6.77}$ | $0.43_{\pm 0.54}$ | $0.80_{\pm 0.08}$ |
| **(e)** W/o environmental reward | $379.91_{\pm 7.87}$ | $\mathbf{0.43_{\pm 0.26}}$ | $0.79_{\pm 0.06}$ |
| EGPO | $\mathbf{388.37_{\pm 10.01}}$ | $0.56_{\pm 0.35}$ | $\mathbf{0.85_{\pm 0.05}}$ |

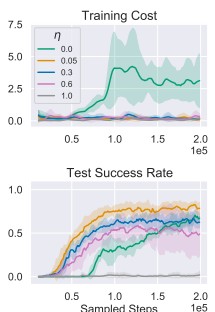

Figure 7: Ablation study on $\eta$.

## 4.3 Ablation Studies

**The impact of expert quality**. To investigate the impact of the expert if its quality is not as good as the well-performing expert used in the main experiments, we involve two expert policies with 60% and 30% test success rate into the training of EGPO. Those two policies are retrieved from the intermediate checkpoints when training the expert. The result of training EGPO with the inferior experts is shown in Fig. 6. We can see that improving the expert's quality can reduce the training cost. This result also empirically justifies the Theorem 1 where the training safety is bounded by the expert safety. Besides, we find better expert leads to better EGPO agent in term of the episodic return. We hypothesize this is because using premature policies as expert will make the switch function produce chaotic intervention signals that mystifies the exploration of the agent.

**The impact of confidence level**. The confidence level $\eta$ is a hyper-parameter. As shown in Fig. 7, we find that when $\eta > 0.05$, the performance decreases as $\eta$ increases. This is because higher $\eta$ means less freedom of free exploration. In the extreme case where $\eta = 1.0$, all data is collected by the expert. In this case, the intervention minimization multiplier $\lambda$ will goes to large value, which damages the training. When $\eta = 0.0$, the whole algorithm reduces to vanilla SAC.

**Ablations of the guardian mechanism.** **(a)** We adopt a rule-based switch designed to validate the effectiveness of the statistical switch in Sec. 3.1. The intervention happens when the distance to the nearby vehicles or to the boundary of road is too small. We find that the statistical switch performs better than rules. This is because it is hard to enumerate manual rules that cover all possible dangerous situations. **(b)** Removing the intervention minimization technique, the takeover frequency becomes extremely high and the agent learns to drive directly toward the boundary of the road. This causes consistent out-of-the-road failures, resulting in the zero success rate and 1 episodic cost. This result shows the importance of the intervention minimization in Sec. 3.3. **(c)** We find that removing the PID controller on updating $\lambda$ in intervention minimization causes a highly unstable training. It is consistent with the result in [6]. We therefore need to use PID controller to optimize $\lambda$ in EGPO and SAC-Lag. **(d)** Removing CQL loss in Eq. 5 damages the performance. We find this ablation reduces the training stability. **(e)** We set the environment reward always to zero in EGPO, so that the only supervision signal to train the policy is the intervention occurrence. This method outperforms IL baselines with a large margin, but remains lower than EGPO in the return and success rate. This suggests EGPO can be turned into a practical online Imitation Learning method.

**Human-in-the-loop experiment**. To demonstrate the potential of EGPO, we conduct a human-in-the-loop experiment, where a human expert supervises the learning progress of the agent. The evaluation result suggests that EGPO can achieve 90% success rate with merely 15,000 environmental steps of training, while SAC-Lag takes 185,000 steps to achieve similar results. EGPO also outperforms Behavior Cloning method in a large margin, while BC even consumes more human data. Please refer to Appendix for more details.

## 5 Conclusion

We develop an Expert Guided Policy Optimization method for the Expert-in-the-loop Reinforcement Learning. The method incorporates the guardian mechanism in the interaction of agent and environment to ensure safe and efficient exploration. The experiments on safe driving show that the proposed method can achieve training and test-time safety and outperform previous safe RL and imitation baselines. In future work we will explore the potential of involving human to provide feedback in the learning process.

**Acknowledgments**

This project was supported by the Centre for Perceptual and Interactive Intelligence (CPII) Ltd under the Innovation and Technology Fund.

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
