# OpenReview forum: "Safe Driving via Expert Guided Policy Optimization"
_robot-learning.org/CoRL/2021/Conference — CoRL2021 Poster_

### Official Review · Reviewer_rHZA · 2021-07-05

**Originality:** Very Good
**Technical Quality:** Good
**Clarity Of Presentation:** Very Good
**Impact:** 3

**Recommendation:**

Weak Accept: I recommend accepting the paper, but will not argue for my recommendation if the majority of other reviewers have a different opinion.

**Summary:**

This paper introduces an expert guided reinforcement learning method, called EGPO, to learn safe driving policies. The expert functions as a safety guardian that will take over the control if the agent’s control action is not safe (i.e., the expert is less likely to take that action). Then the expert control will be learned using offline reinforcement learning technique, conservative Q-learning. To mitigate the problem that the learning agent may totally rely on the expert to take safe actions and do not learn how to drive, the paper introduces a method to minimize intervention using a Lagrangian method. The policy parameters and the Lagrangian multipliers are learned iteratively. In particular, the Lagrangian multipliers are updated using a PID controller.

**Issues:**

Please check the comments above.

There is a typo in (1). It should be if a\in A, choose (a,0); otherwise, choose (a^E,1).

**Reviewer Expertise:**

Very good: Comprehensive knowledge of the area

**Strengths And Weaknesses:**

Strengths:
This paper is well written and easy to follow. The video in the supplementary material greatly helped the reviewer in understanding the evaluation scenarios. The idea of using expert action to directly constraint the action of the learning agent is very interesting. Moreover, the paper explored methods to enhance the learning using offline RL and by minimizing the frequency of expert interference. These methods can be apply to general safe RL algorithms that use safe guardian (e.g., RL + CBF), to enhance their exploration efficiency.

Weaknesses:
- The assumption 1 (that requires the amount of unsafe actions for every state to be bounded in the expert policy) seems too strong. For example, in some unsafe states, it is possible that no action can be safe, in particular, when the action space is constrained. Moreover, it is unclear how the authors can verify this assumption for arbitrary expert policies. In the experiments, the expert is learned using Lagrangian PPO. Does this expert satisfy the assumption? Can this assumption be verified using the success rate (since later the paper uses the success rate the differentiate different expert policies, e.g., in figure 6)?

- In line 231, the paper states that the environmental cost is unavailable to EGPO. What is the motivation of making this assumption? It is understandable that the authors may wish to claim that even without the safety reward directly from the environment, the agent can learn to behave safely with the expert guidance. However, in autonomous driving, this is not a realistic assumption since environmental safety reward is “common sense” and easy to obtain. Moreover, the nature of environmental safety reward is difference from the nature of the expert guidance. The former provides a constraint on the state space, while the latter provides a constraint in the state space. It would be more convincing if the paper demonstrates that the imitation of expert still provides strong performance margin when the environmental safety reward is considered and that how the environmental safety reward affects the amount of expert intervention.

- The authors mention that they plan to extend the method to include human demonstration. How can we obtain a stochastic policy \mathcal{E} in this case?

- In table 1, why is the cumulative cost for the expert higher than that for PPO-Lag for safe RL, offline RL, IL, and the proposed method? For IL and the proposed method, since their safety notions solely comes from the expert demonstration, the results seem counter intuitive. For PPO-Lag in safe RL, is it using the same training method as the expert? Is the difference caused by insufficient training?

- In table 2 (d) and (e), the performance is comparable to EGPO considering the uncertainty range. Does it mean that the proposed method is less sensitive to CQL loss and environmental reward? Why is it the case?

- This paper claims that the proposed method can make learning more efficient. However, the efficiency of the proposed method should also account for the time spent to obtain the expert policy (whether it is a pretrained policy or from human-demonstration).

**Summary Of Recommendation:**

While the paper presents an interesting idea, the major concerns the reviewer has is:
1. the results need more explanation;
2. the assumptions need further justification;
3. how the method can be applied to real world applications, especially how to obtain the demonstration efficiently and how to ensure the expert policy satisfies the assumptions, needs better explanation.

---

> ### Author Response · Authors · 2021-08-26
> **Reviewer rHZA (1/2)**
>
> *(Due to the space limit, we split this response into two parts containing 6 questions and answers. This is the first part.)*
>
> Thank you for the thorough review! Please find the responses below:
>
> ---
>
> *Q1: In line 231, the paper states that the environmental cost is unavailable to EGPO. What is the motivation of making this assumption? The nature of environmental safety reward is different from the nature of expert guidance. The former provides a constraint on the state space, while the latter provides a constraint in the state space.*
>
> A1: We found out this is an error and we will correct this in revision. We actually used the environmental cost as the negative reward and conducted a sort of reward shaping during the experiments.
>
> As shown in the following table, we find that using the safety reward with the original task reward makes marginal differences, compared to ignoring the safety reward.
>
> | Experiment                                  | Training Cost | Test Cost   | Test Reward    | Test Success Rate |
> | ------------------------------------------- | ------------- | ----------- | -------------- | ----------------- |
> | EGPO w/ environment cost as negative reward | 49.95 (4.91)  | 0.56 (0.35) | 388.37 (10.01) | 0.85 (0.05)       |
> | EGPO w/ environment cost discarded          | 46.09 (13.68) | 0.58 (0.20) | 379.42 (21.85) | 0.83 (0.07)       |
>
>
>
> We hypothesize this phenomenon happens because the “intervention minimization” mechanism already imposes the supervision of safety to the learning agent. Therefore the safety reward is redundant and makes little impact on the performance.
>
>
>
> ---
>
> *Q2: The assumption 1 (that requires the amount of unsafe actions for every state to be bounded in the expert policy) seems too strong. For example, in some unsafe states, it is possible that no action can be safe, in particular, when the action space is constrained. Moreover, it is unclear how the authors can verify this assumption for arbitrary expert policies.*
>
> A2: Since the behavior policy is a mixed policy of the learning agent and the expert. It is unlikely to reach a completely dangerous state where no action can be safe.
>
> Our main contribution is on how to better integrate the expert in the trial-and-error exploration in the standard RL diagram to improve the training safety. Therefore we do not expect the training framework can be applied to **arbitrary expert policies**, but instead studying how to exploit the safe expert more efficiently when it is there.
>
>
> ---
>
> *Q3: The authors mention that they plan to extend the method to include human demonstration. How can we obtain a stochastic policy \mathcal{E} in this case?*
>
> A3: We plan to let the human expert intervene and demonstrate directly. Concretely, there is no more takeover function and stochastic expert policy since humans use deterministic policies. Please note that in this case the training workflow is still functional, since we only query an “intervention cost” from the Guardian instead of asking for the action probability during the training.
>
> ---
>
> *Q4: In table 1, why is the cumulative cost for the expert higher than that for PPO-Lag for safe RL, offline RL, IL, and the proposed method? For IL and the proposed method, since their safety notions solely comes from the expert demonstration, the results seem counter intuitive. For PPO-Lag in safe RL, is it using the same training method as the expert? Is the difference caused by insufficient training?*
>
> A4: Please note that the results reported in table 1 is based on the evaluation of the trained agents in test environments. So the “safety notions” do not come from expert demonstration but instead from the trained IL policies in the evaluation.
>
> The success rates of RL, offline RL, IL methods are much worse than the expert. This might result from the fact that we use 100 scenes as the training environments, which might be too difficult for those methods. We find that agents fail to successfully imitate the expert and they drive slowly such that they hardly finish the tasks. However, this also leads to the superior safety performance since no crash would happen at such low speed.

---

> ### Author Response · Authors · 2021-08-26
> **Reviewer rHZA (2/2)**
>
> *(Due to the space limit, we split this response into two parts containing 6 questions and answers. This is the second part.)*
>
>
>
> ---
>
> *Q5: In table 2 (d) and (e), the performance is comparable to EGPO considering the uncertainty range. Does it mean that the proposed method is less sensitive to CQL loss and environmental reward? Why is it the case?*
>
> A5: We find that the introduction of CQL loss and the environmental reward increases the training efficiency and stability greatly.
> The following table shows the test success rate *during the course of training*. Using CQL loss can improve the training efficiency, compared to the EGPO without CQL loss. Interestingly, we find the environmental reward has a marginal impact on the performance, which indicates our framework can run in a reward-free manner, reducing to an online imitation learning method but with better sample efficiency than GAIL and other IL methods.
>
>
> | Experiment: Test Success Rate | 50K steps    | 100K steps   | 150K steps   | 200K steps   |
> | ----------------------------- | ------------ | ------------ | ------------ | ------------ |
> | EGPO                          | 0.272 (0.19) | 0.741 (0.10) | 0.747 (0.12) | 0.852 (0.05) |
> | EGPO w/o CQL loss             | 0.196 (0.16) | 0.657 (0.11) | 0.721 (0.16) | 0.805 (0.08) |
> | EGPO w/o environmental reward | 0.273 (0.19) | 0.720 (0.10) | 0.716 (0.12) | 0.797 (0.06) |
>
>
> ---
>
> *Q6: This paper claims that the proposed method can make learning more efficient. However, the efficiency of the proposed method should also account for the time spent to obtain the expert policy (whether it is a pretrained policy or from human-demonstration).*
>
> A6: Indeed, it is unfair to compare the learning efficiency of our method with RL methods. However, please note that the imitation learning and offline RL methods all require the expert data, while our method needs experts to execute the same amount of steps with the learning agents. We will rephrase the claim and discuss the access of an expert policy and its limitations.

---

> > ### Comment · Reviewer_rHZA · 2021-09-03
> > **Response to Authors' response**
> >
> > I thank the authors for their response and the addition of the human-in-the-loop experiments. I decided to increase my score.

---

### Official Review · Reviewer_hLxx · 2021-07-20

**Originality:** Good
**Technical Quality:** Good
**Clarity Of Presentation:** Good
**Impact:** 3

**Recommendation:**

Weak Accept: I recommend accepting the paper, but will not argue for my recommendation if the majority of other reviewers have a different opinion.

**Summary:**

This paper introduces a method for coupling data collected from an agent with the ones provided by an expert, that can override the actions of the agent in case of unsafe behaviour. The method is tested on an autonomous driving task and outperform a significant amount of baselines.

**Issues:**

- how does your method apply to problems with stochastic dynamics and/or partial observability?
- what do you think your method need to safely handle these problems?
- would the expert be considered as an “oracle” also in noisy partially observable problems? I think this would significantly limit the applicability of the method.

**Reviewer Expertise:**

Very good: Comprehensive knowledge of the area

**Strengths And Weaknesses:**

This paper proposes a method for involving an expert agent in the classical RL loop. The method is theoretically sound, and relatively easy to apply. I like the way the authors addresst the inherent issues of their vanilla method, and the way they propose theoretically grounded solutions to address them, e.g., the use of Conservative Q-Learning and the constrained optimization problem.

I think the paper is overall nice to read, but it has some typos that should be fixed:
- line 141: is a function THAT denotes;
- line 151: negativeLY;
- line 186: stabler -> more stable.

**Summary Of Recommendation:**

The paper introduces a method that I find overall interesting and practical. I find some limitations that I list below that I’d like to be addressed, possibly in the paper, and that motivate my score.

---

> ### Author Response · Authors · 2021-08-26
> **Reviewer hLxx**
>
> Thank you for your review and we are glad that you like this submission.
>
> In this work, we make the clear assumption that an expert is accessible in this expert-in-the-loop RL pipeline. Our main focus is on how to better integrate the expert in the trial-and-error exploration in the standard RL diagram to improve the training safety. Therefore the safe and well-performing expert, which should successfully operate in the environments of interests, should be provided.
>
> Human experts can successfully tackle the challenges such as the stochastic dynamics and partial observability (at least in driving tasks). **Please refer to the common response for a new human-in-the-loop experiment.** Therefore we should not hold concern on the applicability of the method when applying to a human-in-the-loop scheme.
>
> Besides, current experiments already hold partial observability. The learning agent can not access other traffic vehicles’ internal states as well as intentions but can only observe other vehicles’ behavior using self-centered Lidar-like cloud points.

---

> > ### Comment · Reviewer_hLxx · 2021-08-31
> > **Response to Authors' response**
> >
> > I thank the authors for their response that helped me clarify some concerns. After reading the other reviews and respective responses, I decided to keep my score.

---

### Official Review · Reviewer_RZXA · 2021-07-24

**Originality:** Good
**Technical Quality:** Good
**Clarity Of Presentation:** Good
**Impact:** 3

**Recommendation:**

Weak Accept: I recommend accepting the paper, but will not argue for my recommendation if the majority of other reviewers have a different opinion.

**Summary:**

This paper presents an approach for safe policy optimization in the expert-in-the-loop reinforcement learning setting. By assuming that the failure probability of the expert is bounded, the proposed approach shows that the cumulative probability of failure of the trained policy is bounded. Evaluations on a simulator for autonomous driving demonstrates that the approach can achieve high rewards with minimal costs compared to pure imitation learning baselines.

**Issues:**

Please address the issues listed under Weaknesses above (copied below for reference) and a few additional clarifications regarding experiments


- The definition of unsafe actions and states (for example in line 142) is very unclear. In most cases, for example a car that is gradually moving off the road, no single state is safe or unsafe and the trajectory of the car gradually progresses from safe to unsafe regions. So, it is important to clearly define how "unsafety"is quantified in the paper.

- Theorem 1 has a potentially unbounded term. How large can the value of $K^'$ be in Theorem 1? It is important to upper-bound this term because if the term is large, then the bound on cumulative probability of failure becomes vacuous.

- The way the "expert policy" is constructed doesn't seem realistic. In line 248-253, it is mentioned that the expert policy is trained with Lagrangian PPO for a long training time and causes a lot of safety violations during training. This is in a sense circumventing the safety problem because the expert is allowed to be unsafe in training. How will this generalize to a real world experiment? In particular, how will the corresponding expert policy be trained when the approach is transferred to a real-robot navigation task?

- What is the intuition for why offline RL safety violations are much better than Dagger and GAIL but worse than BC? Is the same expert data used to train the IL methods and the offline RL baseline?

- The method presented in the paper did not seem specific to autonomous driving. What are the challenges for applying this method to obtain safe learning policies in locomotion and manipulation tasks? It will be helpful to list out issues, if any.

**Reviewer Expertise:**

Good: General knowledge of the area

**Strengths And Weaknesses:**

Strengths

- The problem of safe policy learning is important and the solution strategy through human-in-the-loop RL is well-motivated. This is a compromise between pure RL methods that do not assume any expert intervention but are unsafe, and pure imitation learning methods that require high quality expert demonstrations and can be much safer in training.

- The theoretical result of bounded training risk is interesting, and important. This guarantee is important to ensure that the safety of the trained behavioral policy will not diverge too much from the safety level of the expert.

Weaknesses

- The definition of unsafe actions and states (for example in line 142) is very unclear. In most cases, for example a car that is gradually moving off the road, no single state is safe or unsafe and the trajectory of the car gradually progresses from safe to unsafe regions. So, it is important to clearly define how "unsafety"is quantified in the paper.

- Theorem 1 has a potentially unbounded term. How large can the value of $K^'$ be in Theorem 1? It is important to upper-bound this term because if the term is large, then the bound on cumulative probability of failure becomes vacuous.

- The way the "expert policy" is constructed doesn't seem realistic. In line 248-253, it is mentioned that the expert policy is trained with Lagrangian PPO for a long training time and causes a lot of safety violations during training. This is in a sense circumventing the safety problem because the expert is allowed to be unsafe in training. How will this generalize to a real world experiment? In particular, how will the corresponding expert policy be trained when the approach is transferred to a real-robot navigation task?

**Summary Of Recommendation:**

My main concern is with the training of the expert policy with a lot of safety violations. As such, the approach doesn't seem scalable to a real world setting, where safety is ultimately important.

---

> ### Author Response · Authors · 2021-08-26
> **Response to Reviewer RZXA (1/3)**
>
> *(Due to the space limit, we split this response into three parts containing 5 questions and answers. This is the first part.)*
>
>
> Thank you for the detailed review! We are glad that you like this submission. Please see the responses below.
>
> ---
>
> *Q1: The way the "expert policy" is constructed doesn't seem realistic. In line 248-253, it is mentioned that the expert policy is trained with Lagrangian PPO for a long training time and causes a lot of safety violations during training. This is in a sense circumventing the safety problem because the expert is allowed to be unsafe in training. How will this generalize to a real world experiment? In particular, how will the corresponding expert policy be trained when the approach is transferred to a real-robot navigation task? My main concern is with the training of the expert policy with a lot of safety violations.*
>
> A1: In this work, we make the clear assumption that an expert is available and accessible in this expert-in-the-loop RL pipeline. Our main contribution is on how to better integrate the expert in the trial-and-error exploration in the standard RL diagram to improve the training safety. This setup is also commonly referred to as human-in-the-loop RL (Christiano et al.) and (Ibarz et al.). Using trained policies to simulate a human expert is a common practice in imitation learning (Ho et al.) and offline reinforcement learning (Fujimoto et al., Fu et al.).
>
> We use PPO-Lagrangian to train the experts because the training algorithm is simple and we can get an expert as an oracle easily. There are alternatives to obtain such experts without violating safety constraints too much, such as imitation learning from human data (Kelly et al.) or curriculum learning for safe RL.
>
> Please note that **our pipeline is agnostic to the expert policy being used**. Using trained policy as the expert is a stepping stone to implement human-in-the-loop RL. As shown in the common response, we introduce the human subject as the expert to monitor the learning process so that the human supervisor takes over the robot once he/she feels it is dangerous.
>
> References:
>
> (Christiano et al.) Deep Reinforcement Learning from Human Preferences
>
> (Ibarz et al.) Reward learning from human preferences and demonstrations in Atari
>
> (Fujimoto et al.) Off-Policy Deep Reinforcement Learning without Exploration
>
> (Fu et al.) D4RL: Datasets for Deep Data-Driven Reinforcement Learning
>
> (Kelly et al.) HG-DAgger: Interactive Imitation Learning with Human Experts
>
> (Ho et al.) Generative Adversarial Imitation Learning

---

> > ### Comment · Reviewer_RZXA · 2021-09-03
> > **Thanks for the response**
> >
> > Thank you for answering my questions in detail. I am satisfied by most of the clarifications, and will lean towards accepting the paper in light of the responses and updated paper.

---

> ### Author Response · Authors · 2021-08-26
> **Response to Reviewer RZXA (2/3)**
>
> *(Due to the space limit, we split this response into three parts containing 5 questions and answers. This is the second part.)*
>
>
> ---
>
> *Q2: The definition of unsafe actions and states (for example in line 142) is very unclear. In most cases, for example a car that is gradually moving off the road, no single state is safe or unsafe and the trajectory of the car gradually progresses from safe to unsafe regions. So, it is important to clearly define how "unsafe" is quantified in the paper.*
>
> A2: Indeed, safety can be defined subjectively or objectively. In a common safe RL setting, a dangerous state is an objective concept determined by the environment, where a cost will occur when agents enter that state. In this work, the environment will also produce the “objective” cost and that is exactly the cost we present in the experiments.
>
> However, minimizing the environmental cost is not enough to ensure the safety of the agents since the danger has already happened when the cost yields. For example, in our environment we will yield cost +1 when the ego vehicle crashes into other vehicles. To eliminate the dangers, one might wish to design an extended cost function that can forecast the danger , for example, yield cost if the vehicle is too close to the obstacles. However, as the case in your example, it is hard to design such logic with handwritten rules, since the designer needs to enumerate all possible cases and also consider other features such as the velocity and inertia of the all vehicles.
>
> Therefore, we turn to follow an alternative way to determine whether a state is safe. Emulating how human drivers judge the situation in practice, we rely on the expert’s own judgement on the safety during training, instead of any external or objective criterion. In Line 142, the “unsafe action that will be rejected by the switch” is exactly the description on how we use a subjective expert-related measurement to determine the safety during training. We will rephase the description to eliminate confusion in revision.
>
> We use the expert-based statistical switch in this work, which can reduce the engineering efforts on designing the rule-based criterions and also lay a stepstone for future human-in-the-loop research where we can replace the expert policy and the switch with a human expert.
>
> To justify the feasibility of the so-called “subjective safety criterion”, we already conducted an experiment where we only activate the expert to take over in unsafe-only situations in **Section 4.3 Ablations of the guardian mechanism (a)**. In that experiment, we define a set of conditions to determine whether a state is safe, such as if the distance to the nearby vehicles or to the road borders is too small etc, and use them as the takeover switch. We copy the results in the table below.
>
> **Please also note that the reported results are evaluated at the test time, when the expert is completely deactivated and the trained agents run in 50 unseen test scenes.**
>
>
> | Experiment                              | Training Cost  | Test Cost   | Test Reward    | Test Success Rate |
> | --------------------------------------- | -------------- | ----------- | -------------- | ----------------- |
> | EGPO w/ rule-based switch               | 127.39 (20.22) | 0.91 (0.60) | 339.10 (11.41) | 0.57 (0.09)       |
> | EGPO w/ expert-based statsitical switch | 73.19 (11.04)  | 0.56 (0.35) | 388.37 (10.01) | 0.85 (0.05)       |
>
> *Note: We add training cost to this table while other metrics are already presented in the original paper.*
>
>
> This experiment shows that the framework works poorly if the takeover function is defined by handwritten objective rules. On the contrary, EGPO, who uses an expert policy and thresholding its confidence to determine the safety “subjectively”, provides better safety and performance during training and testing. The proposed framework not only reduces human efforts to design domain-specified rules, but also enables future human-in-the-loop extension.

---

> ### Author Response · Authors · 2021-08-26
> **Response to Reviewer RZXA (3/3)**
>
> *(Due to the space limit, we split this response into three parts containing 5 questions and answers. This is the third part.)*
>
>
> ---
>
> *Q3: The method presented in the paper did not seem specific to autonomous driving. What are the challenges for applying this method to obtain safe learning policies in locomotion and manipulation tasks? It will be helpful to list out issues, if any.*
>
> A3: We believe the proposed framework is a general training pipeline that incorporates experts into the training loop since we do not introduce any domain-specified designs that limit EGPO to driving domain. This framework therefore can be easily extended to locomotion and manipulation tasks.
>
> However, there is still one challenge if we wish to apply such a method to complex locomotion tasks in the human-in-the-loop setting: what if human beings can not solve the task? For example, if we wish to train an Ant robot with 6 legs, it is hard for a human expert to operate the robot successfully. In that case, a human being can only provide “preference” or “stopping” signals, while no demonstrations can be provided. This case is a future extension of the proposed framework in the human-in-the-loop setting. We will add more discussion on the limitation of the proposed framework in the next version of the paper.
>
> ---
>
> *Q4: What is the intuition for why offline RL safety violations are much better than Dagger and GAIL but worse than BC? Is the same expert data used to train the IL methods and the offline RL baseline?*
>
> A4: It is the same expert data used to train IL and offline RL baselines and the expert is also identical to the one used in EGPO.
>
> We find that CQL indeed is a powerful algorithm that can yield a good solution that achieves better success rate while maintaining lower safety violations. The BC method instead learns a policy that fails to solve the task (so its success rate is low to 57%) and usually conducts noisy actions that lead the vehicle to stop on the road. Therefore the safety violations of BC are lower, in the cost of task performance.
>
> ---
>
> *Q5: Theorem 1 has a potentially unbounded term. How large can the value of K be in Theorem 1? It is important to upper-bound this term because if the term is large, then the bound on cumulative probability of failure becomes vacuous.*
>
> A5: A: $K_\eta$ can not be very large. It refers to the “area” of the “safe action space $A_\eta$”, where all actions will have high probability taken by the expert. Obviously, the area of that part of the action space is bounded by the area of the whole action space. In our experiment, we use a bounded action space $[0, 1]^2$ to denote the steering and acceleration signal to the vehicle.

---

### Official Review · Reviewer_RGdx · 2021-07-24

**Originality:** Good
**Technical Quality:** Good
**Clarity Of Presentation:** Good
**Impact:** 3

**Recommendation:**

Weak Reject: I recommend rejecting the paper, but will not argue for my recommendation if the majority of other reviewers have a different opinion.

**Summary:**

The paper proposes the Export-guided Policy Optimization (EGPO) method aiming to achieve safe learning by combining regular learning with expert policy execution activated using a switch when detecting unsafe situation. The assumption is that a stochastic expert policy exists which can generate a safe action with high probability and also evaluate the probability of taking a given action by the expert. The switch function itself is just a check whether a proposed action has high confidence with respect to the expert model. If not, the expert takes over.

To encourage learning, the frequency of interventions is then limited by adding a term penalizing the excess number of interventions using a Lagrangian multiplier. To improve stability, the multiplier is updated using a PID method.

To train with off-policy expert data, Conservative Q-Learning (CQL) approach is used to improve the training on trajectories with partial expert demonstrations.

The authors perform a study on a simulation autonomous driving task with several type of unsafe situation arising from other vehicles and traffic cones. The results show lower overall cost compared to other methods such as SAC-Lag and PPO-Lag.


other minor comments:

line 52: "the Lagrangian method" is unclear when first introduced
line 151: "negative correlated" is unclear (one needs to read the appendix to understand what this means)
is the policy equation (2) actually used in practice? it is not referenced later in the paper (except once but without concrete application)

**Issues:**

- please answer the list of Weaknesses above

**Reviewer Expertise:**

Very good: Comprehensive knowledge of the area

**Strengths And Weaknesses:**

Strengths:
+ a principled way to integrate expert policy in a learning framework
+ reasonably good performance in a simple simulated environment, compared to other existing methods
+ the authors seem to have also applied the method to more realistic mock-up setting (as shown in Appendix)

Weaknesses:
- it is not very clear whether the method works because the expert is really good and is trained on the exact same environment distribution, and so the learning process is cheating when executing the expert; it would have been interesting to see an expert only being activated in unsafe-only situations (as opposed to just any deviation from the expert policy, which the switch function is based on)
- the usage of Lagrangian multipliers and a PID controller is interesting, but also seem very heuristic and its performance is unclear
- the implications of Theorem 1 are not clear, would be great to refer to it in the actual algorithm, or to use in some constructive way; the term K_eta is also a bit of a mystery, can that term potentially be very large and if so, does this theorem upper bound serve any practical purpose?

**Summary Of Recommendation:**

While integrating an expert policy for safer learning is important, there are some unclear aspects of the approach (as listed in Weaknesses) which need to be addressed before assessing how successful it will be in practice.

---

> ### Author Response · Authors · 2021-08-26
> **Response to Reviewer RGdx (1/2)**
>
> *(Due to space limit, we split this response into two parts containing 5 questions and answers. This is the first part.)*
>
> Thank you for the review! Please see below the responses to the concerns you raised.
>
> ---
>
> *Q1: It is not very clear whether the method works because the expert is really good and is trained on the exact same environment distribution, and so the learning process is cheating when executing the expert; it would have been interesting to see an expert only being activated in unsafe-only situations (as opposed to just any deviation from the expert policy, which the switch function is based on)*
>
> A1: When the expert takes over, it is more like an online demonstration from which the agent can learn and ensure safety. It is not cheating but a feature of our learning-from-demonstration pipeline. This feature is agnostic to the expert being adopted and the expert’s knowledge of the environment.
>
> Since the expert takes over according to the deviation between learner and expert’s policy, there exist two cases: (1) the expert takes over when the learner runs into an unsafe state, (2) the expert takes over in a safe state but the deviation is large. The latter case happens more frequently in the very beginning of training, as shown in Figure 5 of the paper. In this case, the expert will demonstrate professional actions to the agent and so the agent learns quickly. This is an expected behavior and is the key to the high sample efficiency of the proposed method compared to other learning from demonstration methods.
>
> Two experiments presented in the original paper can support the importance of such design.
>
> First, in the **Section 4.3 Ablations of the guardian mechanism (b)**, we present the results when intervention minimization is deactivated. In this experiment, the takeover rate goes to 100% and almost all data is generated by the expert. If executing the expert can lead to high quality data that cheats the training of the learner, then this experiment should yield better performance. However the result turns out to be the worst.
>
> Second, in the **Section 4.3 Ablations of the guardian mechanism (a)**, the expert is only activated in unsafe-only situations defined by a set of handwritten rules, such as if the distance to the nearby vehicles or to the road borders is too small etc. We find that the expert-related statistical switch performs better than rules in test time.
>
> One noticeable result is that the learned policy from EGPO achieves better test performance than the expert in unsafe-only situations. We hypothesize this is due to the case (2) we discussed above, namely the expert provides high quality demonstrations in the beginning of training. Besides, in a practical view, defining the unsafe states with handwritten rules is difficult. Instead, using experts to determine the safety can leverage the foreseeing ability of the expert, which emulates how human drivers judge the potential risky situations.
>
> For your convenience, we copied the relevant results in the following table. **Please also note that the reported results are evaluated on the test set, when the expert is completely deactivated and the trained agents run on  50 unseen test scenes.**
>
> | Experiment                                       | Test Reward    | Test Cost   | Test Success Rate |
> | ------------------------------------------------ | -------------- | ----------- | ----------------- |
> | EGPO w/ 100% takeover rate, Sec 4.3 Ablation (b) | 38.31 (3.61)   | 1.00 (0.00) | 0.00 (0.00)       |
> | EGPO w/ rule-based switch, Sec 4.3 Ablation (a)  | 339.10 (11.41) | 0.91 (0.60) | 0.57 (0.09)       |
> | EGPO                                             | 388.37 (10.01) | 0.56 (0.35) | 0.85 (0.05)       |
>
> *Note: This table is already presented in the original paper.*

---

> ### Author Response · Authors · 2021-08-26
> **Response to Reviewer RGdx (2/2)**
>
> *(Due to space limit, we split this response into two parts containing 5 questions and answers. This is the second part.)*
>
>
> ---
>
> *Q2: The usage of Lagrangian multipliers and a PID controller is interesting, but also seems very heuristic and its performance is unclear.*
>
>
> A2: **We already conducted an experiment where we disable the PID-like updates on Lagrangian multipliers.** The experiment is in **Section 4.3 Ablations of the guardian mechanism (c).** The experimental result suggests that using a PID controller to update the Lagrangian multipliers is crucial to the experiment. For your convenience, we copy the table here to make a clear comparison:
>
>
>
> | Experiment                                 | Test Reward    | Test Cost   | Test Success Rate |
> | ------------------------------------------ | -------------- | ----------- | ----------------- |
> | EGPO                                       | 388.37 (10.01) | 0.56 (0.35) | 0.85 (0.05)       |
> | EGPO w/o PID-like update in SAC-Lagrangian | 338.80 (16.23) | 0.59 (0.40) | 0.67 (0.10)       |
>
> *Note: This table are already presented in the Table 2 of original paper.*
>
>
>
>
>
> The following is a discussion on why the PID controller is essential in our framework. We will add it in the next version.
>
> Lagrangian methods are widely used for constrained optimization problems but show oscillations of the constraints violation during the training. The update of the multiplier, from a perspective of control theory, is an integral control. Introducing a PID controller to update the Lagrangian multiplier can reduce the oscillations and cost violations (Stooke et al.).
>
> In this work, we apply the Lagrangian method in our training to tackle the constrained optimization problem. A challenge is how we incorporate the Lagrangian method with the offline RL algorithm SAC. Though equipping SAC with the Lagrangian method has been proposed in (Ha et al.), the authors do not provide public implementation. We therefore turn to implement the SAC-Lagrangian method on our own and we find that the vanilla SAC-Lagrangian method shows the oscillation issue similar to the one emerged in PPO-Lagrangian (Stooke et al.). As a consequence, we develop the PID SAC-Lagrangian method to alleviate the oscillations. **To the best of our knowledge, this is the first work that combines the PID control of Lagrangian multipliers with an offline RL algorithm.**
>
> (Stooke et al.) Responsive Safety in Reinforcement Learning by PID Lagrangian Methods
>
> (Ha et al.) Learning to Walk in the Real World with Minimal Human Effort
>
>
> ---
>
>
> *Q3: The implications of Theorem 1 are not clear, would be great to refer to it in the actual algorithm, or to use in some constructive way.*
>
> A3: Theorem 1 provides a theoretical guarantee on the system safety during training. It shows that the training safety is bounded by the expert safety. Therefore, if we have an expert being safe enough (for example a human expert), then the system will be safe during training.
>
> ---
>
>
>
> *Q4: The term K_eta is also a bit of a mystery, can that term potentially be very large and if so, does this theorem upper bound serve any practical purpose?*
>
> A4: $K_\eta$ can not be very large. It refers to the “area” of the “safe action space $A_\eta$”, where all actions in it will have high probability taken by the expert. Obviously, the area of that part of the action space is bounded by the area of the whole action space. In our experiments, we use a bounded action space $[0, 1]^2$ to denote the steering and acceleration signal to the vehicle.
>
>
> ---
>
>
> *Q5: Is the policy equation (2) actually used in practice? it is not referenced later in the paper.*
>
> A5: The Eq.2 gives the form of the behavior policy, which is used later in all sampling processes during training. For example, Equation 3 describes the TD error used to train the Q function. We say we use a mixed policy to explore the environment in Line 172, therefore the behavior policy $\pi_\theta(\cdot|s_{t+1})$ in Eq. 3 is replaced by the mixed policy $\hat{\pi}$ in Eq. 2. We also find the behavior policy $\pi_\theta$ in Eq. 6 should be replaced by the mixed policy $\hat{\pi}$. We will revise the paper to remove confusion.

---

### Official Review · Reviewer_o3VH · 2021-07-24

**Originality:** Very Good
**Technical Quality:** Very Good
**Clarity Of Presentation:** Excellent
**Impact:** 4

**Recommendation:**

Strong Accept: I recommend accepting the paper and will argue for my recommendation even if other reviewers hold a different opinion.

**Summary:**

In this paper the authors introduce expert guided policy optimization, an algorithm for safe reinforcement learning. In this method, it is assumed that some expert is available at training time which can output a distribution over expert actions given a state. At its core, the algorithm checks at every stage of training whether an action taken by the "student" will be dangerous or not by checking that it is contained within the confidence interval of the expert's distribution for some threshold nu. The authors then provide an upper bound for the training risk (the expected cumulative probability of failure) using this student-expert scheme. In order to train the student, the authors use a modified version of SAC with a modified loss function for the critic to account for the partial demonstrations provided by the expert. In their experience, the algorithm tends to exploit expert behavior, and therefore, they introduce the intervention cost c, which essentially penalizes expert intervention. This is done by introducing an additional critic which effectively keep track of intervention occurrence, and used as part of a soft constraint in the policy's objective. The authors test the framework on a self-driving environment and are able to obtain better cumulative rewards than similar methods whilst also having low training episodic cost.

**Issues:**

My only concern is that only one environment was used to showcase the algorithm. It would have been nice to have more driving simulators to test the algorithm. Other than that I do not have any other major concerns I would like to raise at this time.

**Reviewer Expertise:**

Fair: Some knowledge of the area

**Strengths And Weaknesses:**

Strengths:
* The paper is very clearly written.
* It provides a realistic experimental setup, even if it is in simulation.

Weaknesses:
* It would have been nice to see the algorithm used in more than one driving simulation environment.

**Summary Of Recommendation:**

I believe this paper proposes an interesting framework for safe reinforcement learning with access to an expert. While there is only one driving environment being showcased, the experiments do provide significant evidence that this method outperforms other similar safe rl methods.

---

> ### Author Response · Authors · 2021-08-26
> **Response to Reviewer o3VH**
>
> Thank you very much for your review and we are glad to see that you like this work! We add a human-in-the-loop experiment to show the effectiveness of the proposed method with a human expert. Please refer to the common response and the revised paper and supplementary material for more information.

---

> ### Comment · Reviewer_o3VH · 2021-08-31
> **Response**
>
> Thanks for the clarifications. I will keep my decision unchanged, albeit my knowledge of this domain is less extensive than other reviewers.

---

### Author Response · Authors · 2021-08-26
**Common Response to All Reviewers**

We are grateful to all reviewers for the detailed reviews!

We revised the paper to address some important issues raised in the reviews. The revision part in the updated PDF is highlighted with dark red color.

Here we would like to emphasize that our main contribution is on integrating the available experts in the trial-and-error exploration in the standard RL diagram to improve the training safety. So we make the clear assumption that an expert is accessible in this expert-in-the-loop RL pipeline.To better motivate the utility of the expert policy, we conduct a new  human-in-the-loop experiment. In this experiment, a human subject acts as the expert to  supervise the learning progress of the agent. The human expert takes over the learning agent once he feels necessary by steering the wheel. In that case, an intervention cost is yielded and the action sequences of the expert are recorded and fed into the replay buffer. All the other parts are the same as the proposed learning pipeline.

The following table shows the experimental result. We find that EGPO with a human expert can achieve a high success rate in merely 15,000 environmental steps, wherein 3911 steps are given by the expert. Meanwhile, SAC-Lagrangian (with PID update) takes 185,000 steps to achieve similar results. We also ask the expert to generate 15,000 steps demonstrations (while in EGPO, only a small part of the 15K steps is given by the expert) and train a BC agent based on those demonstrations. However, BC fails to learn a satisfactory policy.


---


| Experiment                               | Training Cost | Test Reward      | Test Cost     | Test Success Rate |
| ---------------------------------------- | ------------- | ---------------- | ------------- | ----------------- |
| Human expert **(20 episodes)**           | -             | 219.50 (39.53)   | 0.30 (0.550)  | 0.95              |
| Behavior Cloning **(200K steps)**        | -             | 33.21 (5.46)     | 0.990 (0.030) | 0.000 (0.000)     |
| PPO-Lagrangian **(200K steps)**          | 285.1        | 197.76 (7.90)    | 0.427 (0.043) | 0.598 (0.029)     |
| SAC-Lagrangian  **(185K steps)**         | 452.5        | 221.381 (7.90)   | 0.060 (0.049) | 0.940 (0.049)     |
| EGPO (with human expert) **(15K steps)** | 6.14        | 221.058 (32.562) | 0.120 (0.325) | 0.900 (0.300)     |


---

This experiment shows the applicability of the proposed framework with human experts. The demo video of the human-in-the-loop experiment is included in the anonymous Google Drive link:

https://drive.google.com/file/d/1umC7qff_V87zBHdA3nY9WBFgF5GzFXNr/view?usp=sharing

---

### Meta-Review · Area_Chair_6Sx2 · 2021-09-06

**Recommendation:** Accept (Poster)
**Confidence:** 4

**Metareview:**

The authors have sufficiently addressed reviewers' concerns; specifically, the new human-in-the-loop experiment was particularly insightful. I recommend an acceptance.

---

### Decision · Program_Chairs · 2021-09-13

**Decision:**

Accept (Poster)

**Comment:**

The authors have sufficiently addressed reviewers' concerns; specifically, the new human-in-the-loop experiment was particularly insightful. I recommend an acceptance.